# Gradient Estimation For Exactly-$k$ Constraints

## Abstract

The exactly-$k$ constraint is ubiquitous in machine learning and scientific applications, such as ensuring that the sum of electric charges in a neutral atom is zero. However, enforcing such constraints in machine learning models while allowing differentiable learning is challenging. In this work, we aim to provide a "cookbook" for seamlessly incorporating exactly-$k$ constraints into machine learning models by extending a recent gradient estimator from Bernoulli variables to Gaussian and Poisson variables, utilizing constraint probabilities. We show the effectiveness of our proposed gradient estimators in synthetic experiments, and further demonstrate the practical utility of our approach by training neural networks to predict partial charges for metal-organic frameworks, aiding virtual screening in chemistry. Our proposed method not only enhances the capability of learning models but also expands their applicability to a wider range of scientific domains where satisfaction of constraints is crucial.

## 1 Introduction

The exactkly-$k$ constraint, that is, the sum of $n$ variables is equal to $k$, is not only ubiquitous in machine learning such as learning sparse features [Chen et al., 2018] and discrete variational auto-encoders [Rolfe, 2016], but also critical to scientific applications such as charge-neutral scenarios in computational chemistry [Raza et al., 2020] and count-aware cell type deconvolution [Liu et al., 2023]. In the former cases, the variables are binary while in the latter cases, the variables are continuous or integer, depending on the applications. Such tasks can involve optimizing the expectation of an objective function with respect to variables satisfying the exactkly-$k$ constraint, whose distributions are parameterized by neural networks. This optimization problem is challenging since the expectation can be intractable and thus gradient estimation is required. Existing estimators include score-function-based ones that suffer from high variance and reparameterization-based ones that require relaxation and can be highly biased Xie and Ermon [2019]. A recently proposed gradient estimator [Ahmed et al., 2023] outperforms the aforementioned estimators by leveraging constraint probability and avoiding relaxations. Still, it is limited to the exactkly-$k$ constraint on Bernoulli variables.

In this work, we aim to carry out a systematic study of gradient estimation for exactkly-$k$ constraints over Bernoulli, Gaussian, and Poisson variables, the three most commonly used distributions in modeling. We show that on the forward pass, the constrained distributions have closed-form representations, and thus exact sampling from the constrained distribution can be achieved. On the backward pass, we reparameterize the gradient of the loss function with respect to the samples as a function of the expected marginals of the constrained distributions. Further, we find that under certain loss functions, the expected loss under the constrained distribution has a closed-form solution. That is, in such cases, we are able to train models under the exactkly-$k$ constraint without any gradient

Submitted to NeurIPS 2023 AI for Science Workshop.

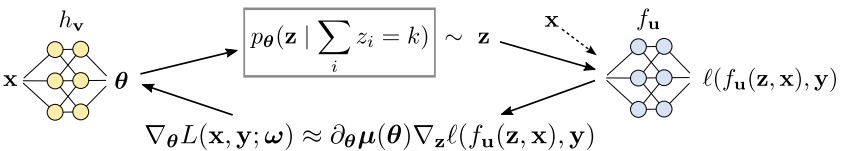

Figure 1: Model formulation under an exactkly-$k$ constraint.

estimations. We include synthetic experiments to evaluate the bias and variance of our proposed gradient estimation on Gaussian and Poisson variables. We also include an experiment on predicting partial charges for metal-organic frameworks, where our gradient estimation, when combined with an ensemble method, achieves state-of-the-art prediction performance.

## 2 Problem Statement and Motivation

We consider models described by the equations

$$\boldsymbol{\theta} = h_{\boldsymbol{v}}(\boldsymbol{x}), \qquad \boldsymbol{z} \sim p_{\boldsymbol{\theta}}(\boldsymbol{z} \mid \sum_i z_i = k), \qquad \hat{\boldsymbol{y}} = f_{\boldsymbol{u}}(\boldsymbol{z}, \boldsymbol{x}), \tag{1}$$

where $\boldsymbol{x} \in \mathcal{X}$ and $\hat{\boldsymbol{y}} \in \mathcal{Y}$ denote feature inputs and target outputs, respectively, $h_{\boldsymbol{v}} : \mathcal{X} \to \Theta$ and $f_{\boldsymbol{u}} : \mathcal{Z} \times \mathcal{X} \to \mathcal{Y}$ are smooth, parameterized maps. $\boldsymbol{\theta}$ are parameters inducing a distribution over the latent vector $\boldsymbol{z}$ and the induced distribution $p_{\boldsymbol{\theta}}(\boldsymbol{z})$ is defined as $p_{\boldsymbol{\theta}}(\boldsymbol{z}) = \prod_{i=1}^n p_{\theta_i}(z_i)$, with $p_{\theta_i}(z_i)$ as defined in Table 1, where $\mathcal{N}(z; \mu, \sigma^2)$ denotes the density of a Gaussian distribution with mean $\mu$ and variance $\sigma^2$ at $z$. An exactkly-$k$ constraint is enforced over the distribution $p_{\boldsymbol{\theta}}(\boldsymbol{z})$, inducing a conditional distribution $p_{\boldsymbol{\theta}}(\boldsymbol{z} \mid \sum_i z_i = k) := p_{\boldsymbol{\theta}}(\boldsymbol{z}) \cdot [\![\sum_i z_i = k]\!]/p_{\boldsymbol{\theta}}(\sum_i z_i = k)$ where the denominator denotes the constraint probability $p_{\boldsymbol{\theta}}(\sum_i z_i = k)$. This formulation is general and it can subsume neural network models that integrate the exactkly-$k$ constraint in the input, output, or latent space, which we visualize in Figure 1.

| VARIABLE | PARAMETERIZED DISTRIBUTION |
|---|---|
| Bernoulli | $p_{\theta_i}(z_i = 1) = \text{sigmoid}(\theta_i)$ $p_{\theta_i}(z_i = 0) = 1 - \text{sigmoid}(\theta_i)$ |
| Gaussian | $p_{\theta_i}(z_i) = \mathcal{N}(z_i; \mu_i, \sigma_i^2)$ with $\theta_i = (\mu_i, \sigma_i)$ |
| Poisson | $p_{\theta_i}(z_i) = \theta_i^{z_i} e^{-\theta_i}/z_i!$ |

Table 1: Parameterization of the three distribution settings.

The training of such models is performed by optimizing an expected loss to learn parameters $\boldsymbol{\omega} = (\boldsymbol{v}, \boldsymbol{u})$ in Equation 1 as below,

$$L(\boldsymbol{x}, \boldsymbol{y}; \boldsymbol{\omega}) = \mathbb{E}_{\boldsymbol{z} \sim p_{\boldsymbol{\theta}}(\boldsymbol{z}|\sum_i z_i = k)}[\ell(f_{\boldsymbol{u}}(\boldsymbol{z}, \boldsymbol{x}), \boldsymbol{y})] \qquad with \ \boldsymbol{\theta} = h_{\boldsymbol{v}}(\boldsymbol{x}), \tag{2}$$

where $\ell : \mathcal{Y} \times \mathcal{Y} \to \mathbb{R}^+$ is a point-wise loss function. However, the standard auto-differentiation can not be directly applied to the expected loss due to two main obstacles. First, for the gradient of $L$ w.r.t. parameters $\boldsymbol{u}$ in the decoder network $f_{\boldsymbol{u}}$ defined as

$$\nabla_{\boldsymbol{u}} L(\boldsymbol{x}, \boldsymbol{y}; \boldsymbol{\omega}) = \mathbb{E}_{\boldsymbol{z} \sim p_{\boldsymbol{\theta}}(\boldsymbol{z}|\sum_i z_i = k)}[\partial_{\boldsymbol{u}} f_{\boldsymbol{u}}(\boldsymbol{z}, \boldsymbol{x})^\top \nabla_{\hat{\boldsymbol{y}}} \ell(\hat{\boldsymbol{y}}, \boldsymbol{y})] \tag{3}$$

with $\hat{y} = f_{\boldsymbol{u}}(\boldsymbol{z}, \boldsymbol{x})$ being decoding of a latent sample $\boldsymbol{z}$, the expectation does not allow closed-form solution in general and requires Monte-Carlo estimations by sampling $\boldsymbol{z}$ from the constrained distribution $p_{\boldsymbol{\theta}}(\boldsymbol{z} \mid \sum_i z_i = k)$. The same issue arises in the gradient of $L$ w.r.t. parameters $\boldsymbol{v}$ in the encoder network defined as

$$\nabla_{\boldsymbol{v}} L(\boldsymbol{x}, \boldsymbol{y}; \boldsymbol{\omega}) = \partial_{\boldsymbol{v}} h_{\boldsymbol{v}}(\boldsymbol{x})^\top \nabla_{\boldsymbol{\theta}} L(\boldsymbol{x}, \boldsymbol{y}; \boldsymbol{\omega}). \tag{4}$$

The second obstacle lies in the computation of the gradient of $L$ w.r.t. the encoder as in Equation 4 defined as $\nabla_{\boldsymbol{\theta}} L(\boldsymbol{x}, \boldsymbol{y}; \boldsymbol{\omega}) := \nabla_{\boldsymbol{\theta}} \mathbb{E}_{\boldsymbol{z} \sim p_{\boldsymbol{\theta}}(\boldsymbol{z}|\sum_i z_i = k)}[\ell(f_{\boldsymbol{u}}(\boldsymbol{z}, \boldsymbol{x}), \hat{\boldsymbol{y}})]$ that requires to compute $\partial_{\boldsymbol{\theta}} \boldsymbol{z}$, a derivative that is not well-defined and requires gradient estimation for updating $\boldsymbol{\theta}$. In a recent work [Ahmed et al., 2023], a gradient estimator called SIMPLE is proposed to tackle these two issues by *exactly sampling* from the constrained distribution and using *marginals* as a proxy to samples respectively, where SIMPLE is able to outperform both score-function-based gradient estimators and reparameterization-based ones. However, SIMPLE is limited to Bernoulli variables and whether the same gradient estimation can be extended to a larger distribution family remains underexplored.

## 3 Gradient Estimation for Exactly-$k$

We tackle the gradient estimation for the exactkly-$k$ constraints by solving the aforementioned two subproblems: **(P1)** how to sample exactly from the constrained distribution $p_{\boldsymbol{\theta}}(\boldsymbol{z} \mid \sum_i z_i = k)$ and **(P2)** how to estimate $\nabla_{\boldsymbol{\theta}} L(\boldsymbol{x}, \boldsymbol{y}; \boldsymbol{\omega})$. By combining solutions to these two problems, we manage to train the constrained model in an end-to-end manner. Table 3 in the Appendix presents a summary of the key components in the proposed gradient estimation.

### 3.1 Exact Sampling

For both Gaussian and Poisson variables, we find that their constrained distributions conform to commonly seen closed-form distributions and thus allow efficient sampling by using built-in sampling algorithms in deep learning frameworks. We formally state our findings below.

**Proposition 1** (Gaussian Constrained Distribution). *Given $\boldsymbol{z} = (z_1, \ldots, z_n)^T$ with $z_i \sim \mathcal{N}(\mu_i, \sigma_i^2)$, the constrained distribution $p(\boldsymbol{z} \mid \sum_{j=1}^n z_j = k)$ is equivalent to an $n-1$ dimensional multivariate normal distribution with mean $\overline{\mu} \in \mathbb{R}^{n-1}$ and covariance matrix $\overline{\boldsymbol{\Sigma}} \in \mathbb{R}^{(n-1) \times (n-1)}$ with their entries defined as below,*

$$\overline{\mu}_i = \sum_{j=1}^{n-1} \left( \mathbb{1}\left[ i = j \right] \sigma_i^2 - \frac{\sigma_i^2 \sigma_j^2}{\sum_{i=1}^n \sigma_i^2} \right) \left( c + \frac{\mu_j}{\sigma_j^2} \right) \quad and \quad \overline{\boldsymbol{\Sigma}}_{i,j} = \begin{cases} \sigma_i^2 - \frac{(\sigma_i^2)^2}{\sum_{i=1}^n \sigma_i^2} & i = j \\ \frac{-\sigma_i^2 \sigma_j^2}{\sum_{i=1}^n \sigma_i^2} & i \neq j \end{cases}.$$

**Proposition 2** (Poisson Constrained Distribution). *Given $\boldsymbol{z} = (z_1, \ldots, z_n)^T$ with $z_i \sim Poisson(\theta_i)$, the constrained distribution $p(\mathbf{z} \mid \sum_{j=1}^n z_n = k)$ is equivalent to a multinomial distribution with parameter $k$ and probabilities $\frac{\theta_1}{\sum_{j=1}^n \theta_j}, \ldots, \frac{\theta_n}{\sum_{j=1}^n \theta_j}$.*

### 3.2 Conditional Marginals as Proxy

For estimating gradient $\nabla_{\boldsymbol{\theta}} L(\boldsymbol{x}, \boldsymbol{y}; \boldsymbol{\omega})$, we follow an approximation adopted by Ahmed et al. [2023], Niepert et al. [2021] where the main intuition is to use the conditional marginals $\boldsymbol{\mu} := \mu(\boldsymbol{\theta}) := \{p_{\boldsymbol{\theta}}(z_j \mid \sum_i z_i = k)\}_{j=1}^n$ as a proxy for samples $\boldsymbol{z}$, that is,

$$\nabla_{\boldsymbol{\theta}} L(\boldsymbol{x}, \boldsymbol{y}; \boldsymbol{\omega}) \approx \partial_{\boldsymbol{\theta}} \mu(\boldsymbol{\theta}) \nabla_{\boldsymbol{z}} \ell(\boldsymbol{x}, \boldsymbol{y}; \boldsymbol{\omega}), \tag{5}$$

where the sample $\boldsymbol{z}$ is reparameterized to be a function of the conditional marginals and is assumed to be $\partial_{\boldsymbol{\mu}} \boldsymbol{z} \approx \mathbf{I}$. In the case of Gaussian and Poisson variables, the reparameterization is achieved by using the expected marginals conditioning on the exactkly-$k$ constraint, that is, $\boldsymbol{\mu} := \mu(\boldsymbol{\theta})$ with $\boldsymbol{\mu}_j = \mathbb{E}_{p_{\boldsymbol{\theta}}(z_j \mid \sum_i z_i = k)}[z_j]$ as a function of the parameters $\boldsymbol{\theta}$. For succinctness, we refer to $\boldsymbol{\mu}$ as expected marginals. The remaining question is how to obtain the expected marginals $\boldsymbol{\mu}$. We find that the expected marginals in both cases have closed-form solutions.

**Proposition 3** (Gaussian Conditional Marginal). *Given $\boldsymbol{z} = (z_1, \ldots, z_n)^T$ with $z_i \sim \mathcal{N}(\mu_i, \sigma_i^2)$, the conditional marginal $p(z_i \mid \sum_{j=1}^n z_j = k)$ follows a univariate Gaussian distribution with mean $\tilde{\mu}_i = \mu_i + \frac{\sigma_i^2}{\sum_{j=1}^n \sigma_j^2}(k - \sum_{j=1}^n \mu_j)$ and variance $\tilde{\sigma}_i^2 = \sigma_i^2 - \frac{(\sigma_i^2)^2}{\sum_{j=1}^n \sigma_j^2}$, that is, $\boldsymbol{\mu}_i = \tilde{\mu}_i$.*

**Proposition 4** (Poisson Conditional Marginal). *Given $\boldsymbol{z} = (z_1, \ldots, z_n)^T$ with $z_i \sim Poisson(\theta_i)$, the conditional marginal of $p(z_i \mid \sum_{j=1}^n z_n = k)$ follows a binomial distribution with parameter $k$ and probability $\frac{\theta_i}{\sum_{j=1}^n \theta_j}$, with $\boldsymbol{\mu}_i = \frac{k\theta_i}{\sum_{j=1}^n \theta_j}$.*

### 3.3 Closed-form Expected Loss

This section focuses on some special cases where the expected loss in Equation 2 has a closed-form solution and thus no gradient estimation is needed. We find that when the decoder $f_{\boldsymbol{u}}$ is an identity function, that is, $\boldsymbol{y} = \boldsymbol{z}$, the expected loss defined over Gaussian variables has a closed-form solution when the element-wise loss is L1 loss or L2 loss. The same conclusion holds for Poisson variables with the element-wise loss being L2 loss. We refer the readers to Proposition 5 and Proposition 6 respectively in Appendix for details.

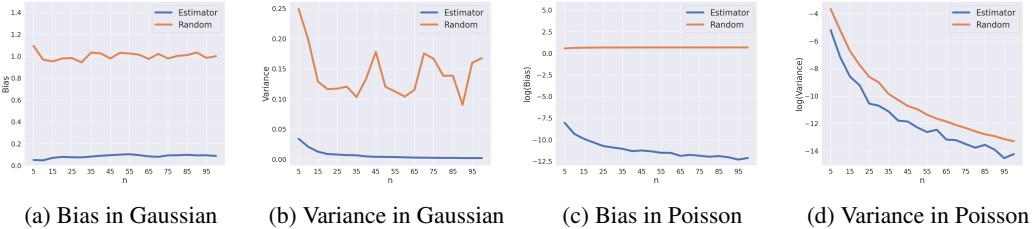

| (a) Bias in Gaussian | (b) Variance in Gaussian | (c) Bias in Poisson | (d) Variance in Poisson |

Figure 2: A comparison of our gradient estimation and random estimations on bias and variance.

## 4 Experiments

We evaluate our proposed gradient estimation on both synthetic settings and a scientific application.

**Synthetic Experiments.** We analyze our proposed gradient estimators for Gaussian and Poisson variables using three metrics, bias, variance, and averaged error, in synthetic settings where the ground truth gradients can be obtained by taking derivatives of the closed-form expected loss as stated in Section 3.3. The distance between the estimated and the ground truth gradient vectors is measured by the cousin distance defined as 1 - cosine similarity. We further compare with a random estimation as a baseline. Bias and variance results are presented in Figure 2 with additional details and results presented in Section C in the Appendix, where our proposed gradient estimator is able to achieve significantly lower bias, variances as well as averaged errors than the baseline, indicating its effectiveness.

**Partial Charge Predictions for Metal-Organic Frameworks.** Metal-organic frameworks (MOFs) represent a class of materials with a wide range of applications in chemistry and materials science. Predicting properties of MOFs, such as partial charges on metal ions, is essential for understanding their reactivity and performance in chemical processes. However, it is challenging due to the complex interactions between metal ions and ligands and the requirement that the predictions need to satisfy the charge neutral constraint, that is, an exactly-zero constraint.

We adopt the same model as in Raza et al. [2020] where the charges are assumed to be Gaussian variables and the element loss is L1 loss, and address this problem by training the model leveraging our observation in Section 3.3 and using gradients of the expected loss. We further observe that using an ensemble of such models gives predictions that also satisfy the charge-neutral constraint. The prediction performance of our two proposed approaches is presented in Table 2, compared with baseline approaches reported by Raza et al. [2020]. Results show that training using closed-form expected loss achieves the same performance as MPNN(variance) which is considered to be the strongest baseline approach, and when further combined with the ensemble method, our approach achieves significantly better predictions.

| METHOD (charge neutrality enforcement) | MAD mean (std) |
|---|---|
| Constant Prediction | 0.324 (7e-3) |
| Element-mean (uniform) | 0.154 (2e-3) |
| Element-mean (variance) | 0.153 (2e-3) |
| MPNN (uniform) | 0.026 (8e-4) |
| MPNN (variance, reproduced) | 0.0251 (8e-4) |
| Closed-form (ours) | 0.0251 (6e-4) |
| Closed-form + Ensemble (ours) | **0.0235 (5e-4)** |

Table 2: Comparison on prediction performance.

## 5 Conclusion

In this work, we provide an extensive study on differentiable learning under exactkly-$k$ constraints given various distribution families. We further provide empirical studies of our proposed gradient estimation on both synthetic experiments and a scientific application.

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

| VARIABLE | SAMPLING | EXPECTED MARGINALS | EXPECTED LOSS |
|---|---|---|---|
| Bernoulli | Proposition 2 in Ahmed et al. [2023] | Theorem 1 in Ahmed et al. [2023] | — |
| Gaussian | Proposition 1 | Proposition 3 | Proposition 5 |
| Poisson | Proposition 2 | Proposition 4 | Proposition 6 |

Table 3: Summary of exact sampling, expected marginals, and closed-form expected loss.

## A  Related Work

A substantial amount of research has been devoted to estimating gradients for categorical random variables. Maddison et al. [2016] Jang et al. [2016] proposed to refactor the non-differentiable sample from a categorical distribution with a differentiable sample from a novel Gumbel-Softmax distribution, which enables automatic differentiation. This paper investigates a more complex distribution, $k$-subset distribution. Gradient estimation under exactkly-$k$ constraints has been widely studied. Existing methods either employ the score function and straight-through estimator or suggest custom relaxation [Kim et al., 2016, Chen et al., 2018, Grover et al., 2019, Xie and Ermon, 2019]. Xie and Ermon [2019] extends the Gumbel-softmax technique to k-subsets, enabling backpropagation for k-subset sampling. However, this comes at the trade-off of introducing some bias in the learning process due to the use of relaxed samples. While score function estimators offer a seemingly simple solution, it is widely acknowledged that they are prone to exhibiting exceedingly high variance. A recently introduced gradient estimator known as SIMPLE [Ahmed et al., 2023] surpasses its predecessors but is constrained to Bernoulli random variables.

Extensive research has been conducted on numerical sampling from multivariate normal distributions while adhering to various constraints. Altmann et al. [2014] reviewed classical Gibbs Sampling on a standard simplex (samples are positive and sum to one) and proposed using Hamiltonian Monte Carlo(HMC) methods. Efficient sampling method for multivariate normal distribution truncated by hyperplanes($\mathbf{Ax} = \mathbf{b}$, where $dim(\mathbf{x}) = N$ and $rank(\mathbf{A}) = n < N$) were investigated by Maatouk et al. [2022] and Cong et al. [2017]. These studies focus on numerical simulations, whereas our approach aims to derive a closed-form solution for the multivariate normal distribution subject to the exactkly-$k$ constraint.

## B  Theoretical Results

**Proposition 5** (Closed-form Expected Loss under Gaussian). *Let $\mathbf{z} = (z_1, \ldots, z_n)^T$, where $z_i \sim \mathcal{N}(\mu_i, \sigma_i^2)$. Let $\mathbf{b} = (b_1, b_2, \ldots, b_n)^T$ be the ground truth vector subject to the equality constraint $\sum_{j=1}^n b_j = k$. The L1 loss of $\mathbf{z}$ subject to the constraint $\sum_{j=1}^n z_j = k$ is given by*

$$L(\theta) = \sum_{i=1}^n \tilde{\sigma}_i \sqrt{\frac{2}{\pi}} \exp\left(\frac{-(\tilde{\mu}_i - b_i)^2}{2\tilde{\sigma}_i^2}\right) + (\tilde{\mu}_i - b_i)\, erf\left(\frac{\tilde{\mu}_i - b_i}{\sqrt{2\tilde{\sigma}_i^2}}\right),$$

*where $\tilde{\mu}_i$ and $\tilde{\sigma}_i^2$ are the mean and variance of the conditional marginal of $z_i$ subject to the constraint. $\tilde{\mu}_i = \mu_i + \frac{\sigma_i^2}{\sum_{j=1}^n \sigma_j^2}(k - \sum_{j=1}^n \mu_j)$ and $\tilde{\sigma}_i^2 = \sigma_i^2 - \frac{(\sigma_i^2)^2}{\sum_{j=1}^n \sigma_j^2}$. Further, the L2 loss of $\mathbf{z}$ subject to the constraint $\sum_{j=1}^n z_j = k$ is given by*

$$L(\theta) = \sum_{i=1}^n \left[\left(\mu_i - \frac{\sigma_i^2 \sum_{j=1}^n \mu_j}{\sum_{j=1}^n \sigma_j^2}\right)^2 + \sigma_i^2 - \frac{(\sigma_i^2)^2}{\sum_{j=1}^n \sigma_j^2} - 2b_i\left(\mu_i - \frac{\sigma_i^2 \sum_{j=1}^n \mu_j}{\sum_{j=1}^n \sigma_j^2}\right) + b_i^2\right].$$

**Proposition 6** (Closed-form Expected Loss under Poisson). *Let $\mathbf{z} = (z_1, \ldots, z_n)^T$, where $z_i \sim Poisson(\theta_i)$. Let $\mathbf{b} = (b_1, b_2, \ldots, b_n)^T$ be the ground truth vector subject to the equality constraint*

$\sum_{j=1}^{n} b_j = k$. *The L2 loss of* $\mathbf{z}$ *subject to the constraint* $\sum_{j=1}^{n} z_j = k$ *is given by*

$$L(\theta) = \sum_{i=1}^{n} \left[ k \left( \frac{\theta_i}{\sum_{j=1}^{n} \theta_j} \right) \left( 1 - \frac{\theta_i}{\sum_{j=1}^{n} \theta_j} \right) + k^2 \left( \frac{\theta_i}{\sum_{j=1}^{n} \theta_j} \right)^2 - 2kb_i \left( \frac{\theta_i}{\sum_{j=1}^{n} \theta_j} \right) + b_i^2 \right].$$

## C  Additional Experiment Results in Synthetic Settings

We carried out a series of experiments to analyze the effectiveness of our gradient estimator from Gaussian and Poisson variables. Our focus lies on three pivotal metrics: bias, variance, and the average error. Since, we only care about the direction of the gradients, we employed the cosine distance, namely 1 - cosine similarity, to measure the deviation of our gradient estimators from the ground truth vector. The ground truth logits, $\mathbf{n}$, are sampled from $\mathcal{N}(\mathbf{0}, \mathbf{I})$ satisfying the constraint. We plotted the three metrics against the dimension of $\mathbf{z}$, namely $n$, and graphed the standard deviations. For each $n$, we randomly generated 10 sets of parameters and calculated the metrics for each set. Then, we take average of these 10 repeats and computed their standard deviations. We compare our results with random guess. The randomly generated gradients are sampled from $\mathcal{N}(\mathbf{0}, \mathbf{I})$.

**Gaussian**  We use the L1 loss function $L(\theta) = \mathbb{E}_{\mathbf{z} \sim p_\theta(\mathbf{z}|\sum_i z_i = 0)}[\| \mathbf{z} - \mathbf{b} \|_1]$. The constraint, $k$, is set to 0. We observe that the bias and average error remain relatively stable across various values of $n$, with biases hovering around 0.1 and average errors hovering around 0.3. The variance steadily decreases and converges to a relatively low value. Our estimator outperforms the baseline across all dimensions in all three metrics.

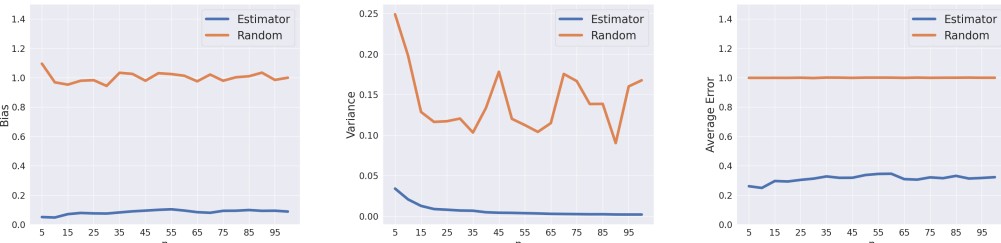

Figure 3: Synthetic Experiment with Gaussian Variables.

**Poisson**  We use the L2 loss function $L(\theta) = \mathbb{E}_{\mathbf{z} \sim p_\theta(\mathbf{z}|\sum_i z_i = 0)}[\| \mathbf{z} - \mathbf{b} \|_2^2]$. The constraint is set to $k = n$. Since, the bias, variance, and average error for our estimators are very small, we opt to take their logarithms. In all dimensions and using all three metrics, our estimator surpasses the baseline.

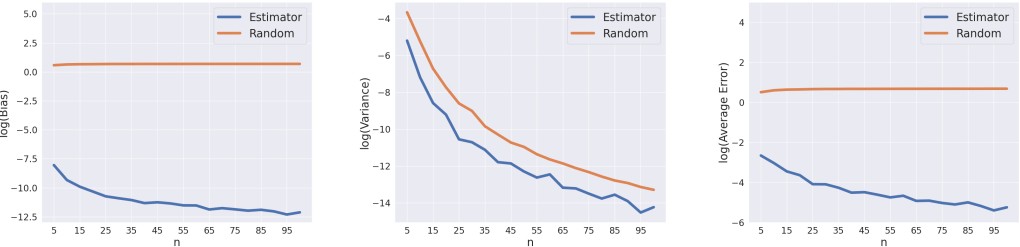

Figure 4: Synthetic Experiment with Poisson Variables.

## D  Additional Experimental Details for Partial Charge Predictions

**Model Architecture**  Our model architecture extends the Message Passing Neural Network (MPNN) Raza et al. [2020] framework and incorporates exact-k constraint for Gaussian variables, ensuring strict adherence to the critical constraint. The core innovation involves replacing the conventional L1 loss with the closed-form Gaussian loss function 5. This loss function penalizes deviations from the exact-k constraint while considering the probabilistic nature of Gaussian variables. This comprehensive approach not only enables our model to capture complex structural relationships in MOFs but also ensures accurate predictions of partial charges while respecting the crucial exact-k constraint, enhancing its applicability in a wide range of graph-based applications, including those pertaining to metal-organic frameworks.

Additionally, we also devise an ensemble methodology to enhance the predictive performance and robustness of our exact-k constrained MPNN model. To achieve this, we adopt a systematic approach encompassing model variability, aggregation strategies and cross-validation. Two instances of the exact-k constrained MPNN model are trained with variations in initialization. We apply the averaging aggregation technique to combine the predictions from these models. Performance assessment is conducted through cross-validation techniques. The ensemble's performance is evaluated on a separate test dataset to ascertain its generalization ability. This ensemble approach not only elevates predictive accuracy but also fortifies the model's resilience, rendering it highly effective for complex tasks, including those pertaining to metal-organic frameworks.

**Training**  Here, we describe our training and evaluation process for the exact-k constrained MPNN. We conducted a random partitioning of the dataset containing 2266 charge-labeled MOFs, creating distinct training, validation, and test sets (70/10/20%). We use the training set for direct model parameter tuning, while the validation set aids in hyperparameter selection to prevent overfitting. The test set plays a crucial role in providing an unbiased assessment of the final model's performance.

**Hyperparameter Tuning**  To optimize our model's performance, we conduct a systematic hyperparameter tuning process, sequentially optimizing six key hyperparameters: Learning rate, Batch size, Time steps, Embedding size, Hidden Feature size, and Patience Threshold. After thorough tuning, we set the hyperparameters to their optimal values: lr = 0.005, batch size = 64, time steps = 4, embedding size = 20, hidden feature size = 40, and patience threshold = 150, achieving peak model performance.

## E  Proofs

### E.1  Proposition 1

*Proof.* Let $\boldsymbol{z} = (z_1, \ldots, z_n)^T$, where $z_i \sim \mathcal{N}(\mu_i, \sigma_i^2)$. We attempt to compute a closed-form solution for the conditional distribution $p\left(\boldsymbol{z} \mid \sum_{j=1}^n z_j = k\right)$.

$$p\left(\mathbf{z} \mid \sum_{j=1}^n z_j = k\right) = \frac{p\left(\boldsymbol{z} \cap \sum_{j=1}^n z_j = k\right)}{p\left(\sum_{j=1}^n z_j = k\right)}$$

$$= \frac{p\left(\boldsymbol{z}\right) \cdot \left[\sum_{j=1}^n z_j = k\right]}{p\left(\sum_{j=1}^n z_j = k\right)}$$

where $\left[\sum z_i = k\right]$ is an indicator function. Notice that the denominator $p(\sum_{j=1}^n z_j = k)$ is the probability distribution function of $Y = \sum_{j=1}^n z_j$ evaluated at $k$. Since $Y$ is a linear combination of independent Gaussian random variables, $Y \sim \mathcal{N}(\sum_{j=1}^n \mu_j, \sum_{j=1}^n \sigma_j^2)$. Thus,

$$p\left(\sum_{j=1}^n z_j = k\right) = \frac{1}{\sqrt{2\pi \sum_{j=1}^n \sigma_j^2}} \exp\left[-\frac{1}{2\sum_{j=1}^n \sigma_j^2}\left(k - \sum_{j=1}^n \mu_j\right)^2\right]$$

The joint distribution function $p(\boldsymbol{z})$, the numerator, follows a multivariate normal distribution with mean $\mu = (\mu_1, \mu_2, \ldots, \mu_n)^T$ and variance $\boldsymbol{\Sigma} = diag\left(\sigma_i^2\right)$ Thus, the conditional distribution can be rewritten as

$$p\left(\boldsymbol{z} \mid \sum_{j=1}^{n} z_j = k\right) = \frac{\prod_{i=1}^{n} \frac{1}{\sqrt{2\pi\sigma_i^2}} \exp\left[-\frac{1}{2\sigma_i^2}(z_i - \mu_i)^2\right]}{\frac{1}{\sqrt{2\pi\sum_{j=1}^{n}\sigma_j^2}} \exp\left[-\frac{1}{2\sum_{j=1}^{n}\sigma_j^2}\left(k - \sum_{j=1}^{n}\mu_j\right)^2\right]}[\sum_{j=1}^{n} z_j = k]$$

Let $C = \left(\frac{1}{\sqrt{2\pi\sum_{j=1}^{n}\sigma_j^2}} \exp\left[-\frac{1}{2\sum_{j=1}^{n}\sigma_j^2}\left(k - \sum_{j=1}^{n}\mu_j\right)^2\right]\right)^{-1}$. We can express our result as

$$p\left(\boldsymbol{z} \mid \sum_{j=1}^{n} z_j = k\right) = C \cdot [\sum_{j=1}^{n} z_j = k] \cdot \prod_{i=1}^{n} \frac{1}{\sqrt{2\pi\sigma_i^2}} \exp\left[-\frac{1}{2\sigma_i^2}(z_i - \mu_i)^2\right]$$

$$= C \cdot f(\boldsymbol{z})$$

where $f(\boldsymbol{z})$ is the joint p.d.f. of the multivariate normal distribution $\boldsymbol{z}$ To deal with the indicator function, let's assume $z_n = k - \sum_{j=1}^{n-1} z_j$. Then, the joint p.d.f. of $\boldsymbol{z}$ becomes

$$f(\boldsymbol{z}) = \frac{1}{\sqrt{2\pi\sigma_n^2}} \exp\left[-\frac{1}{2\sigma_n^2}\left(k - \sum_{i=1}^{n-1} z_i - \mu_n\right)^2\right] \cdot \prod_{i=1}^{n-1} \frac{1}{\sqrt{2\pi\sigma_i^2}} \exp\left[-\frac{1}{2\sigma_i^2}(z_i - \mu_i)^2\right]$$

$$= (2\pi)^{-\frac{n}{2}} \left(\prod_{i=1}^{n} \sigma_i\right)^{-1}$$

$$\exp\left[-\frac{\left(\frac{k^2 - 2k\sum_{i=1}^{n-1} z_i - 2k\mu_n + (\sum_{i=1}^{n-1} z_i)^2 + 2\mu_n\sum_{i=1}^{n-1} z_i + \mu_n^2}{\sigma_n^2} + \sum_{i=1}^{n-1}\frac{z_i^2 - 2z_i\mu_i + \mu_i^2}{\sigma_i^2}\right)}{2}\right]$$

Now, we only consider the terms in the exponential function without $-\frac{1}{2}$.

$$\sum_{i=1}^{n-1}\frac{z_i^2}{\sigma_i^2} + \sum_{i=1}^{n-1}\left(-\frac{2k}{\sigma_n^2} + \frac{2\mu_n}{\sigma_n^2} - \frac{2\mu_i}{\sigma_i^2}\right)z_i + \left(-\frac{2k\mu_n}{\sigma_n^2} + \frac{k^2}{\sigma_n^2} + \frac{\mu_n^2}{\sigma_n^2} + \sum_{i=1}^{n-1}\frac{\mu_i^2}{\sigma_i^2}\right) + \frac{(\sum_{i=1}^{n-1} z_i)^2}{\sigma_n^2}$$

Notice that $(\sum_{i=1}^{n-1} z_i)^2 = \sum_{i=1}^{n-1} z_i^2 + \sum_{i=1}^{n-1}\sum_{j=1,j\neq i}^{n-1} z_i z_j$. Then, our equation becomes

$$\sum_{i=1}^{n-1}\frac{z_i^2}{\sigma_i^2} + \sum_{i=1}^{n-1}\left(-\frac{2k}{\sigma_n^2} + \frac{2\mu_n}{\sigma_n^2} - \frac{2\mu_i}{\sigma_i^2}\right)z_i + \left(-\frac{2k\mu_n}{\sigma_n^2} + \frac{k^2}{\sigma_n^2} + \frac{\mu_n^2}{\sigma_n^2} + \sum_{i=1}^{n-1}\frac{\mu_i^2}{\sigma_i^2}\right)$$

$$+ \frac{\sum_{i=1}^{n-1} z_i^2 + \sum_{i=1}^{n-1}\sum_{j=1,j\neq i}^{n-1} z_i z_j}{\sigma_n^2}$$

$$= \sum_{i=1}^{n-1}\left(\frac{1}{\sigma_i^2} + \frac{1}{\sigma_n^2}\right)z_i^2 + \sum_{i=1}^{n-1}\left(-\frac{2k}{\sigma_n^2} + \frac{2\mu_n}{\sigma_n^2} - \frac{2\mu_i}{\sigma_i^2}\right)z_i + \left(-\frac{2k\mu_n}{\sigma_n^2} + \frac{k^2}{\sigma_n^2} + \frac{\mu_n^2}{\sigma_n^2} + \sum_{i=1}^{n-1}\frac{\mu_i^2}{\sigma_i^2}\right)$$

$$+ \frac{\sum_{i=1}^{n-1}\sum_{j=1,j\neq i}^{n-1} z_i z_j}{\sigma_n^2}$$

$$= \sum_{i=1}^{n-1}\left[\left(\frac{1}{\sigma_i^2} + \frac{1}{\sigma_n^2}\right)z_i^2 + \frac{\sum_{j=1,j\neq i}^{n-1} z_j}{\sigma_n^2}z_i + \left(-\frac{2k}{\sigma_n^2} + \frac{2\mu_n}{\sigma_n^2} - \frac{2\mu_i}{\sigma_i^2}\right)z_i\right]$$

$$+ \left(-\frac{2k\mu_n}{\sigma_n^2} + \frac{k^2}{\sigma_n^2} + \frac{\mu_n^2}{\sigma_n^2} + \sum_{i=1}^{n-1}\frac{\mu_i^2}{\sigma_i^2}\right)$$

Then, we consider an arbitrary $n-1$ dimensional multivariate normal distribution with mean $\overline{\mu}$ and variance $\overline{\boldsymbol{\Sigma}}$. It's p.d.f. is given by

$$(2\pi)^{-\frac{n-1}{2}} \det\overline{\boldsymbol{\Sigma}}^{-\frac{1}{2}} \exp\left(-\frac{1}{2}(\overline{\boldsymbol{z}} - \overline{\mu})^T\overline{\boldsymbol{\Sigma}}^{-1}(\overline{\boldsymbol{z}} - \overline{\mu})\right)$$

We also only consider the terms in the exponential function without $-\frac{1}{2}$. Let $\overline{\mu}_i$ denotes the i-th element of the mean $\overline{\mu}$ and $a_{i,j}$ denotes the i,j-th element of the inverse of the variance and covariance matrix $\overline{\Sigma}^{-1}$.

$$=\overline{z}^T\overline{\Sigma}^{-1}\overline{z} - \overline{z}^T\overline{\Sigma}^{-1}\overline{\mu} - \overline{\mu}^T\overline{\Sigma}^{-1}\overline{z} + \overline{\mu}^T\overline{\Sigma}^{-1}\overline{\mu}$$

$$=\sum_{i=1}^{n-1}\overline{z}_i\left(\sum_{j=1}^{n-1}a_{i,j}\overline{z}_j\right) - \sum_{i=1}^{n-1}\overline{z}_i\left(\sum_{j=1}^{n-1}a_{i,j}\overline{\mu}_j\right) - \sum_{i=1}^{n-1}\overline{\mu}_i\left(\sum_{j=1}^{n-1}a_{i,j}\overline{z}_j\right) + \sum_{i=1}^{n-1}\overline{\mu}_i\left(\sum_{j=1}^{n-1}a_{i,j}\overline{\mu}_j\right)$$

After apply the identity $\sum_{i=1}^{n-1}\overline{z}_i(\sum_{j=1}^{n-1}a_{i,j}\overline{z}_j) = \sum_{i=1}^{n-1}a_{i,i}\overline{z}_i^2 + \sum_{i=1}^{n-1}\sum_{j=1,j\neq i}^{n-1}a_{i,j}\overline{z}_i\overline{z}_j$, the equation becomes

$$=\sum_{i=1}^{n-1}a_{i,i}\overline{z}_i^2 + \sum_{i=1}^{n-1}\sum_{j=1,j\neq i}^{n-1}a_{i,j}\overline{z}_i\overline{z}_j - \sum_{i=1}^{n-1}\overline{z}_i(\sum_{j=1}^{n-1}a_{i,j}\overline{\mu}_j) - \sum_{i=1}^{n-1}\overline{\mu}_i(\sum_{j=1}^{n-1}a_{i,j}\overline{z}_j)$$

$$+ \sum_{i=1}^{n-1}\overline{\mu}_i(\sum_{j=1}^{n-1}a_{i,j}\overline{\mu}_j)$$

$$=\sum_{i=1}^{n-1}\left[a_{i,i}\overline{z}_i^2 + \overline{z}_i\sum_{j=1,j\neq i}^{n-1}a_{i,j}\overline{z}_j - \left(\sum_{j=1}^{n-1}(a_{i,j}+a_{j,i})\overline{\mu}_j\right)\overline{z}_i\right] + \sum_{i=1}^{n-1}\overline{\mu}_i\sum_{j=1}^{n-1}a_{i,j}\overline{\mu}_j$$

Now, we consider the terms in the exponent of this arbitrary $n-1$ dimensional multivariate normal distribution and the $n-1$ dimensional multivariate normal distribution we derived previously.

$$\sum_{i=1}^{n-1}\left[\left(\frac{1}{\sigma_i^2}+\frac{1}{\sigma_n^2}\right)z_i^2 + \frac{\sum_{j=1,j\neq i}^{n-1}z_j}{\sigma_n^2}z_i + \left(-\frac{2k}{\sigma_n^2}+\frac{2\mu_n}{\sigma_n^2}-\frac{2\mu_i}{\sigma_i^2}\right)z_i\right] \tag{6}$$

$$+ \left(-\frac{2k\mu_n}{\sigma_n^2}+\frac{k^2}{\sigma_n^2}+\frac{\mu_n^2}{\sigma_n^2}+\sum_{i=1}^{n-1}\frac{\mu_i^2}{\sigma_i^2}\right)$$

$$\sum_{i=1}^{n-1}\left[a_{i,i}\overline{z}_i^2 + \overline{z}_i\sum_{j=1,j\neq i}^{n-1}a_{i,j}\overline{z}_j - \left(\sum_{j=1}^{n-1}(a_{i,j}+a_{j,i})\overline{\mu}_j\right)\overline{z}_i\right] + \sum_{i=1}^{n-1}\overline{\mu}_i\sum_{j=1}^{n-1}a_{i,j}\overline{\mu}_j \tag{7}$$

Equation (6) is the term in the exponent of an arbitrary $n-1$ dimensional multivariate normal distribution, and Equation (7) is the term in the exponent of previously derived $n-1$ dimensional multivariate normal distribution. We get the following three equations by comparing the first few terms.

$$a_{i,i} = \left(\frac{1}{\sigma_i^2}+\frac{1}{\sigma_n^2}\right) \tag{8}$$

$$a_{i,j} = \frac{1}{\sigma_n^2} \tag{9}$$

$$-\sum_{j=1}^{n-1}(a_{i,j}+a_{j,i})\overline{\mu}_j = \left(-\frac{2k}{\sigma_n^2}+\frac{2\mu_n}{\sigma_n^2}-\frac{2\mu_i}{\sigma_i^2}\right) \tag{10}$$

Equation (8) and (9) define the inverse of the variance and covariance matrix $\overline{\Sigma}^{-1}$. We attempt to compute $\overline{\Sigma}$. Notice that $\overline{\Sigma}^{-1}$ is equivalent to $\mathbf{A}+\mathbf{B}$, where $\mathbf{A} = diag(\frac{1}{\sigma_i^2})$ and every element in matrix $\mathbf{B}$ is $\frac{1}{\sigma_n^2}$.

$$\mathbf{A} = \begin{pmatrix} \frac{1}{\sigma_1^2} & 0 & 0 & \cdots & 0 \\ 0 & \frac{1}{\sigma_2^2} & 0 & \cdots & 0 \\ 0 & 0 & \frac{1}{\sigma_3^2} & \cdots & 0 \\ \vdots & \vdots & \vdots & & \vdots \\ 0 & 0 & 0 & \cdots & \frac{1}{\sigma_{n-1}^2} \end{pmatrix} \quad \mathbf{B} = \begin{pmatrix} \frac{1}{\sigma_n^2} & \frac{1}{\sigma_n^2} & \frac{1}{\sigma_n^2} & \cdots & \frac{1}{\sigma_n^2} \\ \frac{1}{\sigma_n^2} & \frac{1}{\sigma_n^2} & \frac{1}{\sigma_n^2} & \cdots & \frac{1}{\sigma_n^2} \\ \frac{1}{\sigma_n^2} & \frac{1}{\sigma_n^2} & \frac{1}{\sigma_n^2} & \cdots & \frac{1}{\sigma_n^2} \\ \vdots & \vdots & \vdots & & \vdots \\ \frac{1}{\sigma_n^2} & \frac{1}{\sigma_n^2} & \frac{1}{\sigma_n^2} & \cdots & \frac{1}{\sigma_n^2} \end{pmatrix}$$

Consider the following Lemma Miller [1981]

**Lemma 1.** *Let* $\mathbf{G}$ *and* $\mathbf{H}$ *be arbitrary square matrices of the same dimension. If* $\mathbf{G}$ *and* $\mathbf{G} + \mathbf{H}$ *are nonsigular and* $\mathbf{H}$ *has rank one, then*

$$(\mathbf{G} + \mathbf{H})^{-1} = \mathbf{G}^{-1} - \frac{1}{1+g}\mathbf{G}^{-1}\mathbf{H}\mathbf{G}^{-1}$$

*where* $g = tr\left(\mathbf{H}\mathbf{G}^{-1}\right)$

Since $\det \mathbf{A}$ and $\det(\mathbf{A} + \mathbf{B})$ are nonzero, we know that $\mathbf{A}$ and $\mathbf{A} + \mathbf{B}$ are nonsigular. $\mathbf{B}$ is a rank 1 matrix. By the above lemma, we have

$$(\mathbf{A} + \mathbf{B})^{-1} = \mathbf{A}^{-1} - \frac{1}{1+g}\mathbf{A}^{-1}\mathbf{B}\mathbf{A}^{-1}$$

where $g = tr\left(\mathbf{B}\mathbf{A}^{-1}\right)$ This is equivalent to

$$\overline{\Sigma} = \mathbf{A}^{-1} - \frac{1}{1 + tr(\mathbf{B}\mathbf{A}^{-1})}\mathbf{A}^{-1}\mathbf{B}\mathbf{A}^{-1}$$

Equation (6) and (7) imply that $\overline{\Sigma}^{-1}$ is a symmetric and positive definite matrix. Its inverse $\overline{\Sigma}$ is also a symmetric and positive definite matrix. We attempt to find an expression for each element of $\overline{\Sigma}$. We first consider $\mathbf{B}\mathbf{A}^{-1}$.

$$
\mathbf{B}\mathbf{A}^{-1} = \begin{pmatrix}
\frac{1}{\sigma_n^2} & \frac{1}{\sigma_n^2} & \frac{1}{\sigma_n^2} & \cdots & \frac{1}{\sigma_n^2} \\
\frac{1}{\sigma_n^2} & \frac{1}{\sigma_n^2} & \frac{1}{\sigma_n^2} & \cdots & \frac{1}{\sigma_n^2} \\
\frac{1}{\sigma_n^2} & \frac{1}{\sigma_n^2} & \frac{1}{\sigma_n^2} & \cdots & \frac{1}{\sigma_n^2} \\
\vdots & \vdots & \vdots & & \vdots \\
\frac{1}{\sigma_n^2} & \frac{1}{\sigma_n^2} & \frac{1}{\sigma_n^2} & \cdots & \frac{1}{\sigma_n^2}
\end{pmatrix}
\begin{pmatrix}
\sigma_1^2 & 0 & 0 & \cdots & 0 \\
0 & \sigma_2^2 & 0 & \cdots & 0 \\
0 & 0 & \sigma_3^2 & \cdots & 0 \\
\vdots & \vdots & \vdots & & \vdots \\
0 & 0 & 0 & \cdots & \sigma_{n-1}^2
\end{pmatrix}
$$

$$
= \begin{pmatrix}
\frac{\sigma_1^2}{\sigma_n^2} & \frac{\sigma_2^2}{\sigma_n^2} & \frac{\sigma_3^2}{\sigma_n^2} & \cdots & \frac{\sigma_{n-1}^2}{\sigma_n^2} \\
\frac{\sigma_1^2}{\sigma_n^2} & \frac{\sigma_2^2}{\sigma_n^2} & \frac{\sigma_3^2}{\sigma_n^2} & \cdots & \frac{\sigma_{n-1}^2}{\sigma_n^2} \\
\frac{\sigma_1^2}{\sigma_n^2} & \frac{\sigma_2^2}{\sigma_n^2} & \frac{\sigma_3^2}{\sigma_n^2} & \cdots & \frac{\sigma_{n-1}^2}{\sigma_n^2} \\
\vdots & \vdots & \vdots & & \vdots \\
\frac{\sigma_1^2}{\sigma_n^2} & \frac{\sigma_2^2}{\sigma_n^2} & \frac{\sigma_3^2}{\sigma_n^2} & \cdots & \frac{\sigma_{n-1}^2}{\sigma_n^2}
\end{pmatrix}
$$

Notice that $tr(\mathbf{B}\mathbf{A}^{-1}) = \sum_{i=1}^{n-1} \frac{\sigma_i^2}{\sigma_n^2}$, so $1 + tr(\mathbf{B}\mathbf{A}^{-1}) = \sum_{i=1}^{n} \frac{\sigma_i^2}{\sigma_n^2}$. Then we compute $\mathbf{A}^{-1}\mathbf{B}\mathbf{A}^{-1}$

$$
\mathbf{A}^{-1}\mathbf{B}\mathbf{A}^{-1} = \begin{pmatrix}
\sigma_1^2 & 0 & 0 & \cdots & 0 \\
0 & \sigma_2^2 & 0 & \cdots & 0 \\
0 & 0 & \sigma_3^2 & \cdots & 0 \\
\vdots & \vdots & \vdots & & \vdots \\
0 & 0 & 0 & \cdots & \sigma_{n-1}^2
\end{pmatrix}
\begin{pmatrix}
\frac{\sigma_1^2}{\sigma_n^2} & \frac{\sigma_2^2}{\sigma_n^2} & \frac{\sigma_3^2}{\sigma_n^2} & \cdots & \frac{\sigma_{n-1}^2}{\sigma_n^2} \\
\frac{\sigma_1^2}{\sigma_n^2} & \frac{\sigma_2^2}{\sigma_n^2} & \frac{\sigma_3^2}{\sigma_n^2} & \cdots & \frac{\sigma_{n-1}^2}{\sigma_n^2} \\
\frac{\sigma_1^2}{\sigma_n^2} & \frac{\sigma_2^2}{\sigma_n^2} & \frac{\sigma_3^2}{\sigma_n^2} & \cdots & \frac{\sigma_{n-1}^2}{\sigma_n^2} \\
\vdots & \vdots & \vdots & & \vdots \\
\frac{\sigma_1^2}{\sigma_n^2} & \frac{\sigma_2^2}{\sigma_n^2} & \frac{\sigma_3^2}{\sigma_n^2} & \cdots & \frac{\sigma_{n-1}^2}{\sigma_n^2}
\end{pmatrix}
$$

$$
= \begin{pmatrix}
\frac{(\sigma_1^2)^2}{\sigma_n^2} & \frac{\sigma_2^2\sigma_1^2}{\sigma_n^2} & \frac{\sigma_3^2\sigma_1^2}{\sigma_n^2} & \cdots & \frac{\sigma_{n-1}^2\sigma_1^2}{\sigma_n^2} \\
\frac{\sigma_1^2\sigma_2^2}{\sigma_n^2} & \frac{(\sigma_2^2)^2}{\sigma_n^2} & \frac{\sigma_3^2\sigma_2^2}{\sigma_n^2} & \cdots & \frac{\sigma_{n-1}^2\sigma_2^2}{\sigma_n^2} \\
\frac{\sigma_1^2\sigma_3^2}{\sigma_n^2} & \frac{\sigma_2^2\sigma_3^2}{\sigma_n^2} & \frac{(\sigma_3^2)^2}{\sigma_n^2} & \cdots & \frac{\sigma_{n-1}^2\sigma_3^2}{\sigma_n^2} \\
\vdots & \vdots & \vdots & & \vdots \\
\frac{\sigma_1^2\sigma_{n-1}^2}{\sigma_n^2} & \frac{\sigma_2^2\sigma_{n-1}^2}{\sigma_n^2} & \frac{\sigma_3^2\sigma_{n-1}^2}{\sigma_n^2} & \cdots & \frac{(\sigma_{n-1}^2)^2}{\sigma_n^2}
\end{pmatrix}
$$

310    The variance and covariance matrix $\overline{\Sigma}$ becomes

$$\begin{pmatrix} \sigma_1^2 & 0 & 0 & \dots & 0 \\ 0 & \sigma_2^2 & 0 & \dots & 0 \\ 0 & 0 & \sigma_3^2 & \dots & 0 \\ \vdots & \vdots & \vdots & & \vdots \\ 0 & 0 & 0 & \dots & \sigma_{n-1}^2 \end{pmatrix} - \frac{1}{\sum_{i=1}^n \sigma_i^2} \begin{pmatrix} \frac{(\sigma_1^2)^2}{\sigma_n^2} & \frac{\sigma_2^2 \sigma_1^2}{\sigma_n^2} & \frac{\sigma_3^2 \sigma_1^2}{\sigma_n^2} & \dots & \frac{\sigma_{n-1}^2 \sigma_1^2}{\sigma_n^2} \\ \frac{\sigma_1^2 \sigma_2^2}{\sigma_n^2} & \frac{(\sigma_2^2)^2}{\sigma_n^2} & \frac{\sigma_3^2 \sigma_2^2}{\sigma_n^2} & \dots & \frac{\sigma_{n-1}^2 \sigma_2^2}{\sigma_n^2} \\ \frac{\sigma_1^2 \sigma_3^2}{\sigma_n^2} & \frac{\sigma_2^2 \sigma_3^2}{\sigma_n^2} & \frac{(\sigma_3^2)^2}{\sigma_n^2} & \dots & \frac{\sigma_{n-1}^2 \sigma_3^2}{\sigma_n^2} \\ \vdots & \vdots & \vdots & & \vdots \\ \frac{\sigma_1^2 \sigma_{n-1}^2}{\sigma_n^2} & \frac{\sigma_2^2 \sigma_{n-1}^2}{\sigma_n^2} & \frac{\sigma_3^2 \sigma_{n-1}^2}{\sigma_n^2} & \dots & \frac{(\sigma_{n-1}^2)^2}{\sigma_n^2} \end{pmatrix}$$

311    Thus, we have the following result:

$$\overline{\Sigma}_{i,j} = \begin{cases} \sigma_i^2 - \frac{(\sigma_i^2)^2}{\sum_{i=1}^n \sigma_i^2} & i = j \\ -\frac{\sigma_i^2 \sigma_j^2}{\sum_{i=1}^n \sigma_i^2} & i \neq j \end{cases}$$

312    Next, we derive an expression for $\overline{\mu}$. Since $\overline{\Sigma}^{-1}$ is symmetric, Equation (10) can be transformed into

$$-\sum_{j=1}^{n-1} 2a_{i,j} u_j = \left( -\frac{2k}{\sigma_n^2} + \frac{2\mu_n}{\sigma_n^2} - \frac{2\mu_i}{\sigma_i^2} \right)$$

$$\sum_{j=1}^{n-1} a_{i,j} u_j = \left( \frac{k}{\sigma_n^2} + \frac{\mu_i}{\sigma_i^2} - \frac{\mu_n}{\sigma_n^2} \right)$$

313    This is equivalent to

$$\overline{\Sigma}^{-1} \overline{\mu} = c\mathbf{1} + \mu_{\mathbf{reduced}} \oslash \sigma_{\mathbf{reduced}}$$

314    where $c = \frac{k - \mu_n}{\sigma_n^2}$, $\mu_{\mathbf{reduced}} = (\mu_1, \dots, \mu_{n-1})^T$, $\sigma_{\mathbf{reduced}} = (\sigma_1^2, \dots, \sigma_{n-1}^2)^T$, and $\oslash$ denotes
315    element-wise division of vectors. The mean $\mu$ is expressed as

$$\overline{\mu} = \overline{\Sigma}(c\mathbf{1} + \mu_{\mathbf{reduced}} \oslash \sigma_{\mathbf{reduced}}) \tag{11}$$

316    We also attempt to find an element-wise expression for the mean $\overline{\mu}$ Let's define $s_{i.j} = \overline{\Sigma}_{i,j}$. Then we
317    have

$$s_{i,j} = \mathbb{1}\left[ i = j \right] \sigma_i^2 - \frac{\sigma_i^2 \sigma_j^2}{\sum_{i=1}^n \sigma_i^2}$$

318    From the equation for $\overline{\mu}$, we know that

$$\overline{\mu}_i = \sum_{j=1}^{n-1} s_{i,j}\left( c + \frac{\mu_i}{\sigma_i^2} \right)$$

$$= \sum_{j=1}^{n-1} \left( \mathbb{1}\left[ i = j \right] \sigma_i^2 - \frac{\sigma_i^2 \sigma_j^2}{\sum_{i=1}^n \sigma_i^2} \right) \left( c + \frac{\mu_j}{\sigma_j^2} \right)$$

319    Finally, we deal with the constant terms in the exponent.

$$-\frac{2k\mu_n}{\sigma_n^2} + \frac{k^2}{\sigma_n^2} + \frac{\mu_n^2}{\sigma_n^2} + \sum_{i=1}^{n-1} \frac{\mu_i^2}{\sigma_i^2} \tag{12}$$

$$\sum_{i=1}^{n-1} \overline{\mu}_i \sum_{j=1}^{n-1} a_{i,j} \overline{\mu}_j \tag{13}$$

320    Equation (12) is the constant term in the exponential function in the probability distribution function
321    derived by taking the cross section of our $n$ dimensional multivariate normal distribution and a hyper-
322    plane. Equation (13) is the constant term in the exponential function in the probability distribution

function of an arbitrary $n - 1$ dimensional multivariate normal distribution. The scaling term from the exponential term is given by

$$
-\frac{2k\mu_n}{\sigma_n^2} + \frac{k^2}{\sigma_n^2} + \frac{\mu_n^2}{\sigma_n^2} + \sum_{i=1}^{n-1} \frac{\mu_i^2}{\sigma_i^2} - \sum_{i=1}^{n-1} \overline{\mu}_i \sum_{j=1}^{n-1} a_{i,j} \overline{\mu}_j
$$

$$
= \frac{(\mu_n - k)^2}{\sigma_n^2} + \mathbf{1}^T (\mu_{\mathbf{reduced,squared}} \oslash \sigma_{\mathbf{reduced}}) - \overline{\mu}^T \overline{\Sigma}^{-1} \overline{\mu}
$$

where $\mu_{\mathbf{reduced,squared}} = (\mu_1^2, \ldots, \mu_{n-1}^2)^T$. We define

$$
D = \exp\left[ -\frac{1}{2} \left( \frac{(\mu_n - k)^2}{\sigma_n^2} + \mathbf{1}^T \left( \mu_{\mathbf{reduced,squared}} \oslash \sigma_{\mathbf{reduced}} \right) - \overline{\mu}^T \overline{\Sigma}^{-1} \overline{\mu} \right) \right]
$$

This is our scaling term from the exponent. Finally, we consider the constant term in the front.

$$
(2\pi)^{-\frac{n}{2}} \left( \prod_{i=1}^{n} \sigma_i \right)^{-1} = (2\pi)^{-\frac{1}{2}} \frac{\left( \prod_{i=1}^{n} \sigma_i \right)^{-1}}{\det \overline{\Sigma}^{-\frac{1}{2}}} \cdot (2\pi)^{-\frac{n-1}{2}} \det \overline{\Sigma}^{-\frac{1}{2}}
$$

$(2\pi)^{-\frac{n}{2}} \left( \prod_{i=1}^{n} \sigma_i \right)^{-1}$ is the constant term of the multivariate normal truncated by the hyperplane, and $(2\pi)^{-\frac{n-1}{2}} \det \overline{\Sigma}^{-\frac{1}{2}}$ is the constant term of an arbitrary $n - 1$ dimensional multivariate normal. The scaling term is $E = (2\pi)^{-\frac{1}{2}} \frac{\left( \prod_{i=1}^{n} \sigma_i \right)^{-1}}{\det \overline{\Sigma}^{-\frac{1}{2}}}$. Thus, our conditional distribution is a $n - 1$ dimensional multivariate normal distribution with p.d.f. given by

$$
p\left( \mathbf{z} \mid \sum_{j=1}^{n} z_j = k \right) = C \cdot D \cdot E \cdot (2\pi)^{-\frac{n-1}{2}} \det \overline{\Sigma}^{-\frac{1}{2}} \exp\left( -\frac{1}{2} (\overline{\mathbf{z}} - \overline{\mu})^T \overline{\Sigma}^{-1} (\overline{\mathbf{z}} - \overline{\mu}) \right)
$$

where $\overline{\mathbf{z}} = (z_1, \ldots, z_{n-1})^T$. $\qquad\qquad\square$

## E.2  Proposition 2

*Proof.* Let $\mathbf{z} = (z_1, \ldots, z_n)^T$, where $z_i \sim Poisson(\theta_i)$. We attempt to compute a closed-form solution for the conditional probability $p\left( \mathbf{z} \mid \sum_{j=1}^{n} z_j = k \right)$.

$$
p\left( (\mathbf{z} \mid \sum_{j=1}^{n} z_j = k \right) = \frac{p(\mathbf{z} \cap \sum z_i = k)}{p\left( \sum_{j=1}^{n} z_j = k \right)}
$$

Let $Y = \sum_{j=1}^{n} z_j$. The denominator is the p.d.f. of $Y$ evaluated at $k$. Since $Y$ is a linear combination of independent Poisson random variables, we know $Y \sim Poisson(\sum_{j=1}^{n} \theta_j)$. Thus,

$$
p\left( \sum_{j=1}^{n} z_j = k \right) = \frac{e^{-\sum_{j=1}^{n} \theta_j} \left( \sum_{j=1}^{n} \theta_j \right)^k}{k!}
$$

Next, let's consider the numerator.

$$
p(\mathbf{z} \cap \sum_{j=1}^{n} z_j = k) = \begin{cases} p(\mathbf{z}) & \sum_{j=1}^{n} z_j = k \\ 0 & \sum_{j=1}^{n} z_j \neq k \end{cases}
$$

where $p(\boldsymbol{z}) = \prod_{i=1}^{n} f(z_i) = \prod_{i=1}^{n} \frac{e^{-\theta_i}\theta_i^{z_i}}{z_i!}$. Thus, our conditional distribution is given by

$$
p(\mathbf{z}|\sum_{j=1}^{n} z_j = k) = \begin{cases} \frac{\frac{e^{-\sum_{i=1}^{n}\theta_i}\prod_{i=1}^{n}\theta_i^{z_i}}{\prod_{i=1}^{n}z_i!}}{\frac{e^{-\sum_{i=1}^{n}\theta_i}(\sum_{i=1}^{n}\theta_i)^k}{k!}} & \sum_{j=1}^{n} z_j = k \\ 0 & \sum_{j=1}^{n} z_j \neq k \end{cases}
$$

$$
= \begin{cases} \frac{k!\prod_{i=1}^{n}\theta_i^{z_i}}{(\sum_{i=1}^{n}\theta_i)^k \prod_{i=1}^{n}z_i!} & \sum_{j=1}^{n} z_j = k \\ 0 & \sum_{j=1}^{n} z_j \neq k \end{cases}
$$

$$
= \begin{cases} \frac{1}{(\sum_{i=1}^{n}\theta_i)^k} \cdot \frac{k!}{\prod_{i=1}^{n}z_i!} \prod_{i=1}^{n}\theta_i^{z_i} & \sum_{j=1}^{n} z_j = k \\ 0 & \sum_{j=1}^{n} z_j \neq k \end{cases}
$$

$$
= \begin{cases} \frac{k!}{\prod_{i=1}^{n}z_i!} \prod_{i=1}^{n}\left(\frac{\theta_i}{\sum_{j=1}^{n}\theta_j}\right)^{z_i} & \sum_{j=1}^{n} z_j = k \\ 0 & \sum_{j=1}^{n} z_j \neq k \end{cases}
$$

$$
= f\left(\mathbf{z};k,\frac{\theta_1}{\sum_{j=1}^{n}\theta_j},\ldots,\frac{\theta_n}{\sum_{j=1}^{n}\theta_j}\right)
$$

where $f\left(\mathbf{z};k,\frac{\theta_1}{\sum_{j=1}^{n}\theta_j},\ldots,\frac{\theta_n}{\sum_{j=1}^{n}\theta_j}\right)$ is the probability mass function of a multinomial distribution with parameter $k$ and $\frac{\theta_1}{\sum_{j=1}^{n}\theta_j},\ldots,\frac{\theta_n}{\sum_{j=1}^{n}\theta_j}$. $\qquad\square$

### E.3 Proposition 3

*Proof.* Let $\boldsymbol{z} = (z_1,\ldots,z_n)^T$, where $z_i \sim \mathcal{N}(\mu_i,\sigma_i^2)$. We attemp to compute a closed-form solution for the conditional marginal of $z_i$, $p(z_i \mid \sum_{j=1}^{n} z_j = k)$. We first derive the joint distribution of $z_i$ and $\sum_{j=1}^{n} z_j$. Consider the following affine transformation

$$
\mathbf{A}\mathbf{z} = \begin{pmatrix} 0 & \ldots & 1 & \ldots & 0 \\ 1 & \ldots & 1 & \ldots & 1 \end{pmatrix} \begin{pmatrix} z_1 \\ \vdots \\ z_i \\ \vdots \\ z_n \end{pmatrix} = \begin{pmatrix} z_i \\ \sum_{j=1}^{n} z_j \end{pmatrix}
$$

The first row of matrix $\mathbf{A}$ has 1 at i-th column and 0 everywhere, and the last row of matrix $\mathbf{A}$ has 1 everywhere.

**Theorem 2.** *Let* $\mathbf{Y} \sim \mathcal{N}_n(\mu,\boldsymbol{\Sigma})$, *and let* $A$ *be an* $m \times n$ *matrix of rank* $m$. *Then,* $\mathbf{A}\mathbf{Y} \sim \mathcal{N}_m(\mathbf{A}\mu,\mathbf{A}\boldsymbol{\Sigma}\mathbf{A}^T)$ *Gut [2009]*

Since matrix $\mathbf{A}$ is full rank, by Theorem 2, $(z_i,\sum_{j=1}^{n} z_j)^T$ follows a 2 dimensional multivariate normal distribution with mean and variance computed as follows.

$$
\mathbf{A}\mu = \begin{pmatrix} 0 & \ldots & 1 & \ldots & 0 \\ 1 & \ldots & 1 & \ldots & 1 \end{pmatrix} \begin{pmatrix} \mu_1 \\ \vdots \\ \mu_i \\ \vdots \\ \mu_n \end{pmatrix} = \begin{pmatrix} \mu_i \\ \sum_{j=1}^{n} \mu_j \end{pmatrix}
$$

$$
\mathbf{A}\boldsymbol{\Sigma}\mathbf{A}^T = \begin{pmatrix} \sigma_i^2 & \sigma_i^2 \\ \sigma_i^2 & \sum_{j=1}^{n}\sigma_j^2 \end{pmatrix}
$$

**Theorem 3.** *Suppose that* $\mathbf{Y}$, $\mu$, *and* $\boldsymbol{\Sigma}$ *are partitioned as* $\mathbf{Y} = \begin{pmatrix} \mathbf{Y_1} \\ \mathbf{Y_2} \end{pmatrix}$, $\mu = \begin{pmatrix} \mu_1 \\ \mu_2 \end{pmatrix}$, $\boldsymbol{\Sigma} = \begin{pmatrix} \boldsymbol{\Sigma}_{11} & \boldsymbol{\Sigma}_{12} \\ \boldsymbol{\Sigma}_{21} & \boldsymbol{\Sigma}_{22} \end{pmatrix}$, *and* $\mathbf{Y} \sim \mathcal{N}(\mu,\boldsymbol{\Sigma})$. *It can be shown that the conditional distribution of* $\mathbf{Y_1}$ *given* $\mathbf{Y_2}$

353 *is also multivariate normal,* $\mathbf{Y_1} \mid \mathbf{Y_2} \sim N(\mu_{\mathbf{1}|\mathbf{2}}, \mathbf{\Sigma_{1|2}})$*, where* $\mu_{\mathbf{1}|\mathbf{2}} = \mu_{\mathbf{1}} + \mathbf{\Sigma_{12}}\mathbf{\Sigma_{22}}^{-1}(\mathbf{Y_2} - \mu_{\mathbf{2}})$,

354 *and* $\mathbf{\Sigma_{1|2}} = \mathbf{\Sigma_{11}} - \mathbf{\Sigma_{12}}\mathbf{\Sigma_{22}}^{-1}\mathbf{\Sigma_{21}}$ *Holt and Nguyen [2023]*

355 We apply Theorem 3 to derive the conditional distribution. $z_i \mid \sum_{j=1}^{n} z_j \sim \mathcal{N}(\tilde{\mu}_i, \tilde{\sigma}_i^2)$, where the

356 mean and variance are computed as follows:

$$\tilde{\mu}_i = \mu_i + \frac{\sigma_i^2}{\sum_{j=1}^{n} \sigma_j^2}(k - \sum_{j=1}^{n} \mu_j)$$

$$\tilde{\sigma}_i^2 = \sigma_i^2 - \sigma_i^2 \frac{1}{\sum_{j=1}^{n} \sigma_j^2}\sigma_i^2 = \sigma_i^2 - \frac{(\sigma_i^2)^2}{\sum_{j=1}^{n} \sigma_j^2}$$

357 $\qquad\qquad\qquad\qquad\qquad\qquad\qquad\qquad\qquad\qquad\qquad\qquad\qquad\qquad\qquad\qquad\qquad$ $\square$

## E.4 Proposition 4

359 *Proof.* Let $\mathbf{z} = (z_1, \ldots, z_n)^T$, where $z_i \sim Poisson(\theta_i)$. We attempt to compuate a closed-form

360 solution for the conditional marginal $p(z_i \mid \sum_{j=1}^{n} z_n = k)$.

$$p\left(z_i \mid \sum_{j=1}^{n} z_j = k\right) = \sum \cdots \sum_{(z_1,\ldots,z_{i-1},z_{i+1},\ldots,z_n);\sum_{j=1}^{n} z_j = k} p(\mathbf{z} \mid \sum z_i = k)$$

$$= \sum \cdots \sum_{(z_1,\ldots,z_{i-1},z_{i+1},\ldots,z_n);\sum_{j=1}^{n} z_j = k} f\left(\mathbf{z}; k, \frac{\theta_1}{\sum_{j=1}^{n} \theta_j}, \ldots, \frac{\theta_n}{\sum_{j=1}^{n} \theta_j}\right)$$

361 Since the marginal of each variable of a multinomial distribution is a binomial distribution, then the

362 conditional marginal is

$$p\left(z_i \mid \sum_{j=1}^{n} z_j = k\right) = \binom{k}{z_i}\left(\frac{\theta_i}{\sum_{j=1}^{n} \theta_j}\right)^{z_i}\left(1 - \frac{\theta_i}{\sum_{j=1}^{n} \theta_j}\right)^{n-z_i}$$

363 This is the probability mass function of a binomial distribution with parameter $k$ and probability

364 $\frac{\theta_i}{\sum_{j=1}^{n} \theta_j}$. $\qquad\qquad\qquad\qquad\qquad\qquad\qquad\qquad\qquad\qquad\qquad\qquad\qquad\qquad\qquad\qquad$ $\square$

## E.5 Proposition 5

366 *Proof.* Let $\mathbf{z} = (z_1, \ldots, z_n)^T$, where $z_i \sim \mathcal{N}(\mu_i, \sigma_i^2)$. Let $\mathbf{b} = (b_1, b_2, \ldots, b_n)^T$ be the ground

367 truth logits subject to the equality constraint $\sum_{j=1}^{n} b_j = k$. We attempt to derive a closed-form

368 solution for the L1 loss of $\mathbf{z}$ subject to the constraint $\sum_{j=1}^{n} z_j = k$.

$$L(\theta) = \mathbb{E}_{\mathbf{z} \sim p_\theta(\mathbf{z}|\sum_i z_i = 0)}[\|\, \mathbf{z} - \mathbf{b} \,\|_1]$$

$$= \sum_{i=1}^{n} \mathbb{E}_{\mathbf{z} \sim p_\theta(\mathbf{z}|\sum_i z_i = 0)}[\|\, z_i - b_i \,\|_1]$$

369 From previous derivation, we know that the conditional distribution of $z_i$ subject to the equality

370 constraint is an univariate normal distribution wit mean $\tilde{\mu}_i = \mu_i + \frac{\sigma_i^2}{\sum_{j=1}^{n} \sigma_j^2}(k - \sum_{j=1}^{n} \mu_j)$ and

371 variance $\tilde{\sigma}_i^2 = \sigma_i^2 - \frac{(\sigma_i^2)^2}{\sum_{j=1}^{n} \sigma_j^2}$. Let's define $y_i = z_i - b_i$. Then, $y_i \sim N\left(\tilde{\mu}_i - b_i, \tilde{\sigma}_i^2\right)$. Thus,

372 $\mathbb{E}_{\mathbf{z} \sim p_\theta(\mathbf{z}|\sum_i z_i = 0)}[\|\, y_i \,\|]$ is the mean of a folded normal distribution.

$$\sum_{i=1}^{n} \mathbb{E}_{\mathbf{z} \sim p_\theta(\mathbf{z}|\sum_i z_i = 0)}[\|\, y_i \,\|] = \sum_{i=1}^{n} \sigma_{y_i}\sqrt{\frac{2}{\pi}}\exp\left(\frac{-\mu_{y_i}^2}{2\sigma_{y_i}^2}\right) + \mu_{y_i}erf\left(\frac{\mu_{y_i}}{\sqrt{2\sigma_{y_i}^2}}\right)$$

$$= \sum_{i=1}^{n} \overline{\sigma_i}\sqrt{\frac{2}{\pi}}\exp\left(\frac{-(\overline{\mu_i} - b_i)^2}{2\overline{\sigma_i^2}}\right) + (\overline{\mu_i} - b_i)\,erf\left(\frac{\overline{\mu_i} - b_i}{\sqrt{2\overline{\sigma_i^2}}}\right)$$

373 We also attempt to derive a closed-form solution for the L2 loss of $\mathbf{z}$ subject to the constraint
374 $\sum_{j=1}^n z_j = k$.

$$L(\theta) = \mathbb{E}_{\mathbf{z} \sim p_\theta(\mathbf{z} | \sum_i z_i = 0)}[\| \mathbf{z} - \mathbf{b} \|_2^2]$$

$$= \sum_{i=1}^n \mathbb{E}_{\mathbf{z} \sim p_\theta(\mathbf{z} | \sum_i z_i = 0)}[z_i^2] - 2 \sum_{i=1}^n \mathbb{E}_{\mathbf{z} \sim p_\theta(\mathbf{z} | \sum_i z_i = 0)}[z_i b_i] + \sum_{i=1}^n \mathbb{E}_{\mathbf{z} \sim p_\theta(\mathbf{z} | \sum_i z_i = 0)}[b_i^2]$$

375 Since we assume $z_i$ and $b_i$ are independent, and $\mathbf{b}$ is the constant ground truth vector.

$$L(\theta) = \sum_{i=1}^n \mathbb{E}_{\mathbf{z} \sim p_\theta(\mathbf{z} | \sum_i z_i = 0)}[z_i^2] - 2 \sum_{i=1}^n b_i \mathbb{E}_{\mathbf{z} \sim p_\theta(\mathbf{z} | \sum_i z_i = 0)}[z_i] + \sum_{i=1}^n b_i^2$$

$$= \sum_{i=1}^n \mathbb{E}_{z_i \sim p_\theta(z_i | \sum_i z_i = 0)}[z_i^2] - 2 \sum_{i=1}^n b_i \mathbb{E}_{z_i \sim p_\theta(z_i | \sum_i z_i = 0)}[z_i] + \sum_{i=1}^n b_i^2$$

376 From previous derivation, we know that the conditional distribution of $z_i$ is $p\left(z_i \mid \sum_{j=1}^n z_j = k\right) =$
377 $f\left(z_i; \tilde{\mu}_i = \mu_i + \frac{\sigma_i^2}{\sum_{j=1}^n \sigma_j^2}(k - \sum_{j=1}^n \mu_j), \tilde{\sigma}_i^2 = \sigma_i^2 - \frac{(\sigma_i^2)^2}{\sum_{j=1}^n \sigma_j^2}\right)$. The expectation in the first term is
378 the second moment of this gaussian distribution.

$$\sum_{i=1}^n \mathbb{E}_{z_i \sim p(z_i | \sum_i z_i = 0)}[z_i^2] = \sum_{i=1}^n \left[ \left( \mu_i - \frac{\sigma_i^2 \sum_{j=1}^n \mu_j}{\sum_{j=1}^n \sigma_j^2} \right)^2 + \sigma_i^2 - \frac{(\sigma_i^2)^2}{\sum_{j=1}^n \sigma_j^2} \right]$$

379 Likewise, the expectation in the second term is the mean of this gassuain distribution.

$$\sum_{i=1}^n b_i \mathbb{E}_{z_i \sim p_\theta(z_i | \sum_i z_i)} = \sum_{i=1}^n b_i \left( \mu_i - \frac{\sigma_i^2 \sum_{j=1}^n \mu_j}{\sum_{j=1}^n \sigma_j^2} \right)$$

380 $\qquad\qquad\qquad\qquad\qquad\qquad\qquad\qquad\qquad\qquad\qquad\qquad\qquad\qquad\qquad\qquad$ $\square$

## E.6 Proposition 6

382 *Proof.* Let $\mathbf{z} = (z_1, \ldots, z_n)^T$, where $z_i \sim Poisson(\theta_i)$. Let $\mathbf{b} = (b_1, b_2, \ldots, b_n)^T$ be the ground
383 truth vector subject to the equality constraint $\sum_{j=1}^n b_j = k$. We attempt to derive a closed-form
384 solution for the L2 loss of $\mathbf{z}$ subject to the constraint $\sum_{j=1}^n z_j = k$.

$$L(\theta) = \mathbb{E}_{\mathbf{z} \sim p_\theta(\mathbf{z} | \sum_i z_i = 0)}[\| \mathbf{z} - \mathbf{b} \|_2^2]$$

$$= \sum_{i=1}^n \mathbb{E}_{z_i \sim p_\theta(z_i | \sum_j z_j = 0)}[z_i^2] - 2 \sum_{i=1}^n b_i \mathbb{E}_{z_i \sim p_\theta(z_i | \sum_j z_j = 0)}[z_i] + \sum_{i=1}^n b_i^2$$

385 Since the conditional marginal distribution is a binomial distribution, it's second moment is given by

$$\sum_{i=1}^n \mathbb{E}_{z_i \sim p_\theta(z_i | \sum_j z_j = 0)}[z_i^2] = \sum_{i=1}^n \left[ k \left( \frac{\theta_i}{\sum_{j=1}^n \theta_j} \right) \left( 1 - \frac{\theta_i}{\sum_{j=1}^n \theta_j} \right) + k^2 \left( \frac{\theta_i}{\sum_{j=1}^n \theta_j} \right)^2 \right]$$

386 It's first moment(mean) is given by

$$-2 \sum_{i=1}^n b_i \mathbb{E}_{z_i \sim p_\theta(z_i | \sum_j z_j = 0)}[z_i] = -2k \sum_{i=1}^n b_i \left( \frac{\theta_i}{\sum_{j=1}^n \theta_j} \right)$$

387 Thus, we have

$$\sum_{i=1}^n \mathbb{E}_{z_i \sim p_\theta(z_i | \sum_j z_j = 0)}[z_i^2] = \sum_{i=1}^n \left[ k \left( \frac{\theta_i}{\sum_{j=1}^n \theta_j} \right) \left( 1 - \frac{\theta_i}{\sum_{j=1}^n \theta_j} \right) + k^2 \left( \frac{\theta_i}{\sum_{j=1}^n \theta_j} \right)^2 \right]$$

$$-2k \sum_{i=1}^n b_i \left( \frac{\theta_i}{\sum_{j=1}^n \theta_j} \right) + \sum_{i=1}^n b_i^2$$

388 $\qquad\qquad\qquad\qquad\qquad\qquad\qquad\qquad\qquad\qquad\qquad\qquad\qquad\qquad\qquad\qquad$ $\square$

