# OpenReview forum: "Gradient Estimation For Exactly-$k$ Constraints"
_NeurIPS.cc/2023/Workshop/AI4Science — NeurIPS2023-AI4Science Poster_

### Official Review · Reviewer_6XVb · 2023-10-16
**A good estimator for exactly-k constraint gradient**

**Rating:** 7
**Confidence:** 4

**Review:**

The author proposes theoretical and experimental analysis for gradient estimation under exactly-k constraint. The authors propose closed-form estimation for some cases, and the analysis shows that the estimator outperforms the empirical SOTA when leveraging ensemble techniques.

In general, the paper is in good shape, and is worth to be published. Some suggestions I would like to propose:
(i) The author may provide sensitivity analysis to provide more evidence about the comparison to the SOTA.
(ii) The author may provide computation burden (resources, time, etc) for each methods to better show if the new estimator can beat SOTA by achieving SOTA more efficiently.
(iii) The author may provide another discussion section in appendix to clearly view their contribution to the empirical side, and point out their future step, including the potential applications that might be suitable for this estimator

---

### Official Review · Reviewer_E8X4 · 2023-10-20

**Rating:** 7
**Confidence:** 3

**Review:**

This paper provides a nice method of Gradient Estimation For Exactly-k Constraints. The only question I have is: the algorithm seems generic, the paper could discussion other applications where this method can be applied.